# Edge Effect in Electronic and Transport Properties of 1D Fluorinated Graphene Materials

**DOI:** 10.3390/nano12010125

**Published:** 2021-12-30

**Authors:** Jingjing Shao, Beate Paulus

**Affiliations:** Institut für Chemie und Biochemie, Freie Universität Berlin, Arnimallee 22, 14195 Berlin, Germany; b.paulus@fu-berlin.de

**Keywords:** graphene, transport, local current density, fluorination, edge effect, spintronics

## Abstract

A systematic examination of the electronic and transport properties of 1D fluorine-saturated zigzag graphene nanoribbons (ZGNRs) is presented in this article. One publication (Withers et al., *Nano Lett.*, **2011**, *11*, 3912–3916.) reported a controlled synthesis of fluorinated graphene via an electron beam, where the correlation between the conductivity of the resulting materials and the width of the fluorinated area is revealed. In order to understand the detailed transport mechanism, edge-fluorinated ZGNRs with different widths and fluorination degrees are investigated. Periodic density functional theory (DFT) is employed to determine their thermodynamic stabilities and electronic structures. The associated transport models of the selected structures are subsequently constructed. The combination of a non-equilibrium Green’s function (NEGF) and a standard Landauer equation is applied to investigate the global transport properties, such as the total current-bias voltage dependence. By projecting the corresponding lesser Green’s function on the atomic orbital basis and their spatial derivatives, the local current density maps of the selected systems are calculated. Our results suggest that specific fluorination patterns and fluorination degrees have significant impacts on conductivity. The conjugated π system is the dominate electron flux migration pathway, and the edge effect of the ZGNRs can be well observed in the local transport properties. In addition, with an asymmetric fluorination pattern, one can trigger spin-dependent transport properties, which shows its great potential for spintronics applications.

## 1. Introduction

Due to the special sp${}^{2}$ hybridized electrons in graphene [1], the 2D allotrope of carbon has an enormous amount of potential applications in optoelectronics, such as spintronics [2,3,4]. In particular, the ultra-fast electron mobility in graphene makes it a promising new-generation transistor material. The lack of a bandgap, however, leads to an unsatisfying ON/OFF ratio of the on-state current and off-state current in the device, which is the most essential characteristic in electronic transistor applications. Via heteroatom functionalizations, the electronic, magnetic, and optical properties of graphene can be modulated, which extends its potential range of applications. Fluorine chemistry belongs to one of the most efficient approaches to introducing a suitable bandgap [5,6,7,8] in the system. Many previous studies [9,10,11,12] have reported on the synthesis of 2D fluorinated graphene materials. Depending on the exposure duration of the fluorination reactants, the stoichiometry of the carbon and fluorine atoms of the material can be tuned accordingly. Although atomically precise controlled fluorination is experimentally challenging, several successful syntheses have been reported [13,14,15]. These resulting materials have a wide range of electronic properties, from metallic to insulating [11,16,17,18,19]. One specific synthesis of fluorinated graphene [13] has been reported, where the width of the fluorinated area in pristine graphene can be well controlled via electron beam-scanning the material in a row-like fashion. The experimental result has also revealed that the conductivity of such a material is strongly influenced by the size of the fluorinated area.

In addition to 2D fluorinated graphene, most recent experiments have shown the feasibility of 1D fluorinated graphene nanoribbons (GNRs) [20,21]. GNRs [22,23,24], the most generic form of graphene nano structures, are potential candidates for nano wires with tailored conductance properties. Depending on the edge shapes, GNRs can be divided into two major groups: arm-chair GNRs (AGNRs) and zigzag GNRs (ZGNRs). With the modulation of the width of the material, their electronic structures vary significantly. The well-known edge effect of ZGNRs, where two spin projections (channels) near the Fermi level occupy opposite edges, holds great promise for the possible application in spintronics [25,26].

Inspired by this preliminary knowledge, we investigate 1D ZGNRs-based transport models with various fluorination degrees. In the considered structures, the sp${}^{2}$ dangling bonds at the zigzag edges are saturated with fluorines, and the fluorination occurs at the sp${}^{2}$ hybridized carbon atoms in the GNRs sheet. The advantage of the 1D structure resides in terms of the computational cost of the transport calculations. As for the simulation of the 2D structures, both different super-cell structures and extra boundary conditions [27] need to be considered, which is more cumbersome than using the 1D structure for the transport models. In addition, the transport events, from the source to the drain, dominantly occur in one dimension, and we are confident that the 1D transport model can represent the main features occurring in the experimentally synthesized 2D materials [28,29]. The NEGF is one of the most frequently employed methodologies for obtaining the transport properties at the quasi-stationary limit [30,31,32]. It is often combined with a DFT Hamiltonian, which is based on the assumption that ground-state DFT provides a good approximation of the current-carrying scattering states in a non-equilibrium transport process [33]. Although the combination has many limitations and restrictions, such as being unable to describe the excited states or the dynamics of the system, including spin-flipping and spin dephasing [34], a strong correlation regime, as well as the failure to capture the transient phenomena, it is still currently the most practical way of obtaining insights based on atomistic modelling with an outstanding balance of efficiency and accuracy [35,36,37,38,39,40]. In addition to the density functional theory (DFT) + non-equilibrium Green’s function (NEGF) combination, our recent implementation [41] on local current density maps has been already demonstrated as an efficient procedure for gaining detailed information about the local transport properties, such as the electron flux migration pathways. Moreover, the spin-resolved version offers the insight of the spin characteristics in the ZGNRs. In order to simulate different sizes of the fluorinated area in the the scattering region, two factors are tuned: (1) the width of the ZGNRs, and (2) the fluorination degrees. This set-up is thought to mimic the experimentally realized materials [13]. In our best hope, a systematic study of the electronic and transport properties of the considered structures could offer insights into the detailed structure–function relation, which potentially assists the design of new fluorinated graphene-based transistor materials.

## 2. Model Construction, Methods, and Computational Details

### 2.1. Model Construction

A model system, composed of a scattering region connected with two semi-infinite electrodes, is shown in Figure 1. The Hamiltonians of the complete system are partitioned into three Hamilton matrix blocks: $H_{\mathrm{scat}}$, $H_{L}$, and $H_{R}$. A detailed description of the Hamilton matrix construction procedure can be found in our previous publication [41]. All transport models used in this work are composed of 24 units of ZGNRs. The scattering region (12 units) and the L/R electrodes (6 units) in each model have the same chemical composition, which minimises the possible geometric distortion in the whole structure. The convergence of the length of each segment is chosen in such a way that the local energy spectra of the scattering region and of the electrodes are aligned with each other (See Appendix A), which ensures the periodic nature of the complete system.

### 2.2. Methods

The general expression of the transmission function calculated from the Green’s function expression is
(1)$$T\left(E\right)=Tr[G^{r}\left(E\right)\Gamma_{L}\left(E\right)G^{a}\left(E\right)\Gamma_{R}\left(E\right)].$$

With the Green’s function, it is
(2)$$G\left(z\right)={\left(zS_{\mathrm{scat}}-H_{\mathrm{scat}}-\Sigma_{L}\left(z\right)-\Sigma_{R}\left(z\right)\right)}^{-1},$$
where $H_{\mathrm{scat}}$ and $S_{\mathrm{scat}}$ are the Hamiltonian and overlap matrices of the scattering region. $\Sigma_{L}/R\left(z\right)$ are the self-energies describing the effect of the semi-infinite electrodes on the scattering region. The detailed procedure of constructing the required matrices and their definitions can be found in our previous publication [41]. The retarded and advanced Green’s function $G^{r/a}\left(E\right)$ in the transmission function formalism Equation (Equation 1) are obtained by using, respectively, $z=E\pm i0^{+}$. It is noted that the transmission function for each spin channel is calculated individually. From the transmission function, the zero-voltage conductance is obtained from the expression $\frac{2e^{2}}{h}T\left(E_{F}\right)$, where *e*, *h*, and $E_{F}$ are, respectively, the elementary charge, Planck’s constant, and the Fermi energy of the system. The spectral broadening matrices in the transmission function are given by $\Gamma_{L}/R=i\left(\Sigma_{L/R}^{r}-\Sigma_{L/R}^{a}\right)$, which accounts for the level of broadening in the scattering region due to the coupling to the electrodes. Here, we wish to focus on the GNRs-scattering region itself, putting aside the possible complications due to the electronic effect of the metal contacts linked to the reservoir and the electrodes. In [42,43], a simple model of the metallic electrodes without substantial electronic features is suggested: semi-infinite GNRs with highly broadened states, which effectively smear out the bandgap inside of the electrodes. It has also been found that the broadened transmission function is consistent with the experimental value [44]. Therefore, for the same purpose, we have adapted their solution and added a finite numerical imaginary part η to the $z=E\pm i\eta^{+}$ in the self-energy calculations of the electrodes. This is also known as the ’wide-band limit’ approach [45,46], which assumes that the detailed structure of the density of states in the electrodes is not important for the description of transport. The transmission function obtained from different η values (0, 0.1, 0.5, 1 eV) for F-6ZGNRs is presented in Appendix A, where 0.5 eV and 1 eV both give the converged transmission function. Therefore, 0.5 eV is chosen to apply for all following transport calculations.

When a bias voltage is applied between the electrodes, $\left(-e\right)V_{\mathrm{voltage}}=\mu_{R}-\mu_{L}$, the total electric current per spin channel is given by
(3)$$I_{e}=\frac{-e}{h}\int_{-\infty }^{+\infty }T\left(E\right)\left(f_{L}\left(E\right)-f_{R}\left(E\right)\right)\mathrm{dE},$$
where the $f_{L/R}\left(E\right)$ are the Fermi distribution functions of the left and right contacts. It is noted that the effect of the bias voltage on the system is not considered, as the electric field effect is small in a sufficiently long system. It is, thus, reasonable to use the zero-voltage electronic structure to simplify the device-electrode interfaces [43].

For the current density calculation, we follow the approach developed in [47,48,49]. The non-equilibrium Keldysh Green’s function $G^{<}\left(E\right)$ (lesser Green’s function) is calculated from the advanced and retarded Green’s function as:(4)$$G^{<}\left(E\right)=iG^{r}\left(E\right)\left[f_{L}\left(E\right)\Gamma_{L}+f_{R}\left(E\right)\Gamma_{R}\right]G^{a}\left(E\right).$$

In order to obtain a real-space representation of the local currents, the lesser Green’s function is expanded using the atomic orbitals ($\psi_{\mu_{A}/\nu_{B}}\left(r\right)$) at position r in the real-space representation:(5)$$G^{<}\left(r,r^{\prime },E\right)=\sum_{A,B}\sum_{\mu_{A}\nu_{B}}\psi_{\mu_{A}}\left(r\right)G_{\mu_{A}\nu_{B}}^{<}\left(E\right)\psi_{\nu_{B}}\left(r^{\prime }\right).$$

The current density per spin can be represented as a spatial derivative of the Keldysh function:(6)$$\begin{array}{cc} j(r,E) & =\frac{1}{2\pi }\frac{\hbar }{2m}\lim_{r^{\prime }\to r}\left(\nabla_{r^{\prime }}-\nabla_{r}\right)G_{\sigma }^{<}\left(r,r^{\prime },E\right)\\ \; & =\frac{1}{2\pi }\frac{\hbar }{m}\sum_{A,B}\sum_{\mu_{A}\nu_{B}}\psi_{\mu_{A}}\left(r\right)G^{as}\nabla \psi_{\nu_{B}}\left(r\right), \end{array}$$
where $G^{\mathrm{as}}$ is an abbreviation for the antisymmetric elements of the lesser Green’s function $\frac{1}{2}\left(G_{\mu_{A}\nu_{B}}^{<}-G_{\nu_{B}\mu_{A}}^{<}\right)$.

In our previous work, we have introduced two numerical techniques to evaluate Equation (Equation 6) with improved computational efficiency: a real-space filter using the compressed row storage (CRS) format [50], and a spectral filter based on single value decomposition (SVD). A detailed description can be found in the Method section in [51]. Here, we apply the CRS on the atomic orbitals-based $\psi_{\mu_{A}}\left(r\right)$ matrix and their derivatives, and the SVD technique on the matrix $G^{\mathrm{as}}$ containing the antisymmetric elements. Additionally, the local current density takes the form of a reduced diagonal matrix of singular value $\Sigma_{red}$, with the selected $N_{\mathrm{red}}$ prominent values of Σ,
(7)$$\begin{array}{cc} j(r,E) & =\frac{1}{2\pi }\frac{\hbar }{m}\Psi^{T}\left(r\right)U_{red}\Sigma_{red}V_{red}^{T}\nabla \Psi \left(r\right)\\ \; & =\frac{1}{2\pi }\frac{\hbar }{m}\Phi^{T}\left(r\right)\Sigma_{red}\left(\nabla \Phi \right)\left(r\right), \end{array}$$
where the matrices $\Phi^{T}\left(r\right)$ and $\left(\nabla \Phi \right)\left(r\right)$ are obtained by linear transformations using the rectangular matrices $\left\{U_{red},V_{red}\right\}$ of the singular vectors, i.e.,
(8)$$\begin{array}{cc} \Phi^{T}\left(r\right) & =\Psi^{T}\left(r\right)U_{red} \end{array}$$
(9)$$\begin{array}{cc} \left(\nabla \Phi \right)\left(r\right) & ={V_{red}}^{T}\nabla \Psi \left(r\right). \end{array}$$

The total local current is obtained by the integration of the local current density over the energy window for each spin channel, which is defined by applied voltage V.
(10)$$J\left(r\right)=\int_{\frac{-eV}{2}}^{\frac{eV}{2}}j\left(r,E\right)\mathrm{dE}.$$

### 2.3. Computational Details

To determine the first principle values for the Hamiltonian matrices elements in Figure 1, spin-polarized DFT calculations for both periodic electrodes (6 units) and the single Γ-point of the considered transport model (24 units in total) are performed using the GPAW [52,53] package. For the electrodes, the computational cell is sampled using 24 Monkhorst-Pack K-points along the periodic direction. The structure relaxations employ the LibXC [54,55] implementation of the PBE functional [56]. The wave functions are represented using a numerical double-ζ polarized (dzp) basis set [57]. During the structure optimization, atomic positions are varied until the remaining forces are less than 0.02 eV/Å. The density matrix is integrated using a Fermi–Dirac distribution with a $k_{B}T$ value of 0.0001 eV, and the wave functions are represented with a grid spacing of 0.18 Å. The electronic transport calculations are done via the atomic simulation environment (ASE) implementation of the NEGF formalism [58,59,60,61,62]. The projection of the local current density on a grid is processed using the ORBKIT [63,64,65] package.

## 3. Results and Discussions

### 3.1. Hydrogen-Saturated ZGNRs and Fluorine-Saturated ZGNRs

The bandgaps of 2–13 widths of ZGNRs with both hydrogen- (denoted as H-ZGNRs) and fluorine-edge saturation (denoted as F-ZGNRs) are shown in Figure 2. Comparing ZGNRs with the same width, H-ZGNRs have, in general, larger bandgaps than F-saturated ones. The increase in the width results in a similar trend in both type of ZGNRs. Starting with 2ZGNRs, a relatively low band gap (0.35 eV for F-2ZGNRs and 0.45 eV for H-2ZGNRs) is obtained. From 3ZGNRs to 6ZGNRs, F-ZGNRs have an around 0.45 eV bandgap, while H-ZGNRs have bandgaps between 0.50–0.55 eV. From 6ZGNRs to 12ZGNRs, a constant decrease in the bandgaps is observed, where the bandgap consistently decreases as an increasing width is observed, converging to 0.30 eV for F-ZGNRs and 0.33 eV for H-ZGNRs. Due to the narrow width of the 2ZGNRs, the local magnetic moments on the edge carbon atoms (0.054 $\mu_{b}$) is much smaller in comparison to the wider ZGNRs, such as 4ZGNRs and 6ZGNRs (0.083 $\mu_{b}$). This reflects that carbon atoms in between the edges are influenced by the fluorine atoms from the opposite edges, which, we believe, is the physical reason behind the non-monotonical changes.

In Figure 3, the bandstructures and PDOS on two different groups of carbon atoms of 6ZGNRs are shown. Similar electronic structures are obtained from both 6ZGNRs. The α and β spin channels are located at the opposite edge positions, although the PBE bandstructures of the two spin channels appear identical. This observation is the well-known ’edge effect’.

Both H-ZNGRs and F-ZGNRs show similar behaviours and only small quantitative differences in their electronic structures. Therefore, we have only considered F-ZGNRs in the following. This selection is supported by the harsh conditions of the fluorination reactions with graphitic materials. It is reasonable to assume that all dangling bonds are saturated with fluorine instead of with hydrogen, and this assumption is consistent with the finding that the C–H bond, e.g., in benzene, is weaker than the C–F bond [66].

### 3.2. Fluorination Pattern on Fluorine-Saturated ZGNRs

In this section, we examine the impact of different fluorination degrees on both thermodynamic stabilities and electronic structures. In the experiment [13], the fluorination area can be well controlled by the electron beam so that the materials are fluorinated in a row-like fashion. In order to simulate this special characteristic, we use F-6ZGNRs to examine the effect of different fluorination degrees and configurations. In Figure 4, the top views of all simulated structures are illustrated. For both the (2) 33.33% and (4) 66.67% cases, two different configurations are considered—the (A) asymmetric and the (B) symmetric configurations. Asymmetric configurations cause more structural bulking in the *z*-axis plane of the ZGNRs, whereas the symmetric fluorination patterns lead to relatively flat structures. In Table 1, the bandgap for the two spin channels and the associated binding energies for these structures are listed. The total binding energy (E${}_{a}$) is obtained as the energy difference between the sum of the total number of F${}_{2}$ plus the pristine F-ZGNRs, and the total energy of the optimized partially fluorinated F-ZGNRs. For a better comparison, the total binding energy is converted into an average binding energy per F${}_{2}$ (E${}_{a}$/F${}_{2}$). Symmetric fluorination configurations (B) show higher thermodynamic stabilities in comparison to the asymmetric ones, as more energy is required to deform the flat ZGNRs in the latter. At all degrees, the fluorination is highly exothermic, with slight changes in the reaction energy depending on the actual fluorination degree and pattern, where the asymmetric configuration is approximately 0.2 eV higher in energy than the symmetric ones. A similar trend has also been observed in 2D fluorinated graphene materials [67].

Pristine F-6ZGNRs have a bandgap of 0.45 eV, which can be changed drastically via different fluorination degrees. Asymmetric configurations with lower fluorination degrees have relatively small bandgaps in one spin channel, but relatively large ones in the other, such as 16.67% (2(A)) and 33.33% (3). The symmetric configurations, on the contrary, neither change the bandgap of pristine F-ZGNR significantly nor break the spin degeneracy in the bandstructures. In Figure 5, the electronic structures and HOCO/LUCO of 2(A)/2(B) and 4(A)/4(B) are shown. The ones of the other partially fluorinated structures are shown in Appendix A. In 16.67% of asymmetric fluorinated F-6ZGNRs, the band structures of α and β spin channels differ drastically from each other, in contrast to symmetric patterns, which exhibit similar electronic structures as the pristine F-6ZGNR. The edge effect is visible for these two configurations. Interestingly, with the increase in the fluorination degree, the difference between the two spin channels disappears, e.g., in the 66.67% case. Although the HOCO/LUCO are located at different carbons in (4B), they belong to the same conjugated π system.

Combining the considerations of the width and the fluorination degrees, a systematic study of fluorine-saturated 6ZGNRs/8ZGNRs/12ZGNRs, with symmetric fluorination patterns starting from the edges, is conducted. The structures of all considered F-8ZGNRs and F-12ZGNRs can be found in Appendix A. For the F-12ZGNRs cases, different orientations of the F${}_{2}$ are also considered (Appendix A). We found that, overall, the total energies and bandgaps do not differ much between different orientations of F${}_{2}$.

It is noted that, although the PBE functional usually underestimates the bandgap values severely [68,69], they can still provide a first indication for the feasibility of constructing electronic devices.

### 3.3. Transport Properties on Selected Models

The experimentally measured conductivity [13] of the materials is largely determined by the width of the remaining pristine graphene area, i.e., the conductivity decreases with the increasing fluorination degree. In order to gain deep insights into the conductivity properties, we construct the transport models of the selected structures to investigate their transport properties. For simplification, the F-*X*ZGNRs-based transport model (24 units) is abbreviated as F-*X*Z in the following, where *X* denotes the width.

#### 3.3.1. The Width of the F-ZGNRs

The first factor we want to consider is the impact of the width of the F-ZGNRs on conductivity. Due to the large computational demand, a systematic investigation is completed for the F-*X*Z transport model, where X = 2–8 and 12. The total current-bias voltage dependences of these systems are shown in Figure 6. It is noted that the results present both α and β channels. The results for odd numbers of *X* can be found in Appendix A (Appendix A). They follow a similar trend as the even-numbered F-*X*Zs, and provide no further information; we therefore focus our discussion on the even-numbered F-*X*Zs.

From 0 V to 0.4 V, both F-2Z and F-12Z have relatively large conductivities in comparison to the other systems. At 0.3 V, there is a strong increase for the F-12Z, indicating the opening of a new conductive channel in the system. For F-4Z and F-6Z, a large increase can be found at 0.4 V. At 0.5 V, both F-2Z and F-12Z conduct the largest amount of total current, followed by F-8Z, F-4Z, and F-6Z. With a further increase of the applied bias voltage, ohmic behaviours are observed in all F-*X*Zs. At 1 V, F-12Z obtains the largest conductivity, and both F-4Z and F-6Z have similar values, around 25 μA.

These behaviours can all be understood in relation to their electronic structures. Although the HOCO and LUCO of the two spin channels are located at opposite edges, they possess the same bandgap; therefore, the same total current-bias voltage dependence is observed. This argument is also supported by the local energy spectra (Appendix A), where the same energy levels are found for two spin channels. The amount of the applied bias voltages inducing a strong increase in the conductivity for the systems is approximately equal to the bandgap values, where 0.3 V for F-12Z and 0.4 V for F-8Z, F-4Z, and F-6Z are needed. Since the direct bandgap does not locate exactly at the Γ point in the F-*X*ZGNRs, while the Γ-point calculations for the transport models are used to approximate the periodic nature of the system, the exact correlation between the conductivity and the bandgap is not expected. When the applied bias voltage is increased above 0.6 V, the amount of total conducted current in the systems follows the same trend as their bandgaps. This could be understood as evidence that when the applied bias voltage is large enough, the conductive states from the local energy spectra (Appendix A) near the Fermi level can be well populated. The extra states, which originate from the Γ-point approach, no longer affect the system; thus, the trend of the bandgaps in the IV dependence is recovered.

In Figure 7, the local current density maps for the scattering region of F-6Z for both the α and β spin channels at two different applied bias voltages are illustrated. Overall, the electron flux takes the pathways along the lateral C–C bonds in the same direction as the applied bias voltage through the system. From the side view, the majority of the current density is observed above and under the ZGNRs plane. This indicates that the conjugated π systems perpendicular to the nanostructure plane are dominantly responsible for the conductance. When we compare the α and β spin channels at 0.5 V (the upper two panels), a higher current density is found at the edge of the F-6Z, and two spin channels occupy the opposite sites, which coincides very well to the LUCO for each spin channel shown in Figure 3. In the third panel, the local current maps of both the α and β spin channels appear the same at 1 V, i.e., when the applied bias voltage is increased, the difference between the two spin channels disappears and the spin degeneracy is recovered. It can also be explained from the perspective of the electronic structures, where the orbitals far away from the Fermi level are spin-degenerate, and no edge effect is observed beyond 0.5 eV away from the Fermi level. The current density maps also explain the strong increase at 0.5 V in the total current-bias voltage dependence: since all conjugated π channels in the structures start to conduct electrons, the significant increase in the conductivity is inevitable.

#### 3.3.2. Fluorination Degrees of the F-ZGNRs

The next factor we want to consider is the influence of various fluorination degrees on conductivity. We, thus, investigate the transport properties of fluorinated F-6ZGNRs-based structures (see Figure 4 and Appendix A). The corresponding local energy spectra of the scattering regions can be found in Appendix A.

In Figure 6, the total current-bias voltage dependence of asymmetric structures with five different fluorination degrees is shown. In general, asymmetric configurations with lower fluorination degrees have higher conductivity in the α channel than the β channel, whereas both spin channels conduct the same amount of current in the structures with 66.67% and 83.33% fluorination degrees. As discussed above, we know that the α channels of the structures with fluorination degrees between 16.67% and 50% have almost zero bandgaps. These characteristics are reflected in their conductivities, where clear ohmic behaviours are observed in the α channels, while their β spin channels behave very differently under the same applied bias voltage. In the β channels, the total current is almost zero until 0.5 V, and at 0.7 V, a further increase is observed. This feature corresponds well to their bandgaps around 0.7 eV. The structure with the 66.67% fluorination degree has a 0.184 eV bandgap, but it starts to conduct current already at very low bias voltages. This might be explained with the size of unfluorinated π system. As shown in Figure 8, F-2Z has an unexpectedly high conductivity, and it has the same number of remaining conjugated π systems as the 66.67% fluorinated F-6Z. When the fluorination degree is increased to 83.33%, no conjugated π system is available any more, which leads to a much lower conductivity, even at higher bias voltages.

For the symmetric configurations, we have included the structures derived from both F-6ZGNRs and F-8ZGNRs. The corresponding local energy spectra of the scattering regions can be found in Appendix A. As shown in Table 2, this type of fluorination pattern does not change the bandgap of F-*X*ZGNRs drastically, which is also observed in their global transport properties as shown in Figure 9. A strong increase in the conductivity is found at 0.4 V for all symmetric fluorinated F-6Zs and F-8Zs. Both spin channels, in all cases, conduct the same amount of the total current. In comparison, the structures derived from F-8Zs conduct slightly more current than the F-6Zs ones, as there are more conjugated π channels available in F-8Zs. These observations are all in line with their electronic structures, and the trend of the observation at 1 V corresponds well to the bandgaps listed in Table 2.

From the total current-bias voltage dependence, we learn that spin channels behave very differently depending on the fluorination pattern, especially with lower fluorination patterns, such as 16.67%, 33.33%, and 50%. Although the two spin channels of both configurations with a 66.67% fluorination degree behave the same in the global transport properties, asymmetric configurations can conduct double the amount of the total current, compared to the symmetric ones. In order to have more insight into their local transport properties, in Figure 10 and Figure 11, the local current density maps, and both the asymmetric and symmetric flourination configurations of the 33.33% and 66.67% fluorinated F-6Zs, are shown.

In the asymmetric configuration with the 33.33% fluorination degree, the α and the β spin channel have very similar current density patterns, where the difference is barely recognizable. However, in the integrated property—the total current—the difference between the two spin channels can be clearly observed (Figure 6). The local current density is the atomistic resolution of the integrated total current. Since the variation in the current density magnitude is much larger than in the IV curve, the difference between α and β in this case is rather insignificant. The conjugated π system near the fluorination area conducts the most amount of current in the α channel. The upper-edge C–C bond is the unfavourable pathway, which corresponds to the LUCO shape shown in Figure 5. When the applied bias voltage is increased to 1 V, the magnitude of the current density increases and the migration path of the electrons remains unchanged. It is also found that more electron flux populates the middle region of the C–C bonds with a higher applied bias voltage, and some eddy currents are observed in the connecting area between the conjugated π system and the C–F bond, which is very likely caused by the structural bulking.

In the symmetric structure, at 0.4 V, the electrons flow dominantly at the one side of the ZGNRs plan of the α spin channel in the 33.33% symmetric fluorinated F-6Z, whereas the β spin channel is at the other side. With the increase in the bias voltage, an increased current density is observed and more current travels along the unfluorinated C–C bonds. The migration path of the electron flux is congruent with the LUCOs shown in the Figure 5, where the edge effect is retained with the symmetric fluorination pattern.

In both configurations of the 66.67% fluorinated F-6Z, there is no difference between the two spin channels; therefore, only the α spin channel is illustrated. The spin degeneracy in the conductance is supported by the LUCOs shown in Figure 5. The remaining conjugated π system in these two configurations is the only migration path of the electron flux passing through the junction. The LUCOs of the two spin channels are both located within this conjugated π system. Geometrically, the fluorinated area in the asymmetric configuration is located at one side of the ZGNRs, whereas, in the symmetric configuration, the fluorinated area is distributed evenly at both sides. This feature leads to more current density residues populated near the edge of the fluorinated area in the symmetric structures than in the asymmetric ones. Moreover, the majority of the residues are scattered away from the junction, instead of passing through the junction, which, eventually, does not contribute to the transferred current. This explains why asymmetrically fluorinated F-ZGNRs are more conductive than symmetric ones at 1 V.

## 4. Conclusions

In this contribution, we have presented a systematic investigation on the electronic and transport properties of edge-fluorinated ZGNRs with various widths and fluorination degrees. The specific row-like fluorination pattern is thought to mimic the geometrically well-controlled experimental materials via electron beam-scanning the material in a row-like fashion.

It is found that both the width of the F-ZGNRs and the fluorination degrees have an impact on the bandgaps of the system. Interesting features, such as the edge effect, is reflected as opposite edge occupations of the spin orbitals. The symmetric fluorination pattern of the F-ZGNRs does not change the electronic structure drastically. The asymmetric fluorination pattern, however, results in very different behaviours of the spin channels.

We subsequently constructed transport models based on selected F-*X*ZGNRs to study their global and local current properties. In general, a strong increase in the total current is found at a bias voltage that approximately coincides with their bandgap values. The trend of the conductivity at high bias voltages correlates to the trend of the bandgaps, as all conductive states near the Fermi level can be well populated with a large bias voltage. From the local current density maps, we observe that the electron flux travels through the junction mainly via a conjugated π system. In pristine F-6Zs, a relatively high current density at the edge carbons at low bias voltages is found. With the increase of the applied bias voltage, the difference in the conductance between the spin channels disappears.In both configurations of the 33.33% fluorinated F-6Zs, the electron flux migration path has its preferable side of conjugated π systems. With a further increase of the fluorination degree, the edge effect diminishes in significance, especially when there is only one conjugated π system remaining. The migration pathways of the electron flux at low bias voltages can all be well justified by the shape of the LUCOs of the corresponding F-*X*ZGNRs. Overall, our transport calculations have revealed the essential consequences of the edge states of the corresponding *X*ZGNRs on their local conductivity.

These findings suggest that, with the modification of a conjugated π system, the conductivity of the systems can be tuned drastically. Our results show that, in general, the conductivity of the ZGNRs increases with the extent of the π system. One conjugated π system, as an exception, can conduct much more current than expected. At the atomistic simulation level, although the width is an important factor for the conductivity of the structure, the exact fluorination degrees and fluorination patterns have a more determined influence. The local current map analysis provides detailed information on the conductivity of the system, and the resulting electron flux migration pathways coincide with the LUCOs of the corresponding F-*X*ZGNRs.

To this end, 1D fluorine-saturated ZGNRs are shown to be very promising materials for spintronics applications, due to their the edge effect-induced spin-dependent conductivity.

Finally, as a word of caution, the effect caused by the substrate, defects, and rising temperature, in principle, should have a strong effect on the system. These aspects deserve entirely new studies and, therefore, should be the subject of future investigations.

## Figures and Tables

**Figure 1 nanomaterials-12-00125-f001:**
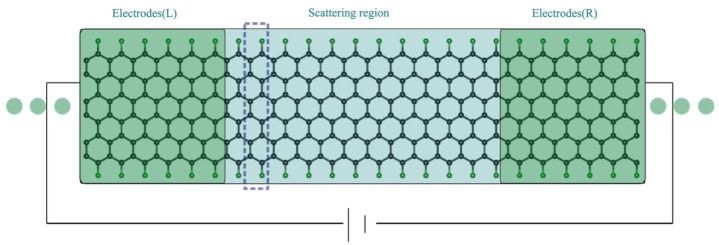
Representation of a transport model based on 6ZGNRs with fluorinated edges. Carbon atoms are drawn in black, the fluorine atoms are in green. One ZGNR unit is marked with dash lines. The nanojunction is partitioned into three parts: the central scattering region (12 units) and two electrodes parts (L/R with 6 units). The Hamiltonians and the overlapping matrices of the complete system are partitioned into three Hamilton matrix blocks: $H/S_{\mathrm{scat}}$, $H/S_{L}$, and $H/S_{R}$, respectively. A detailed description of the Hamilton matrix construction procedure can be found in our previous publication [41]. The electrodes part of the transport Hamiltonians, shown in the green boxes, is set with periodic boundary conditions, which satisfies the semi-infinite feature requirement in the partition procedure of the complete Hamiltonian system.

**Figure 2 nanomaterials-12-00125-f002:**
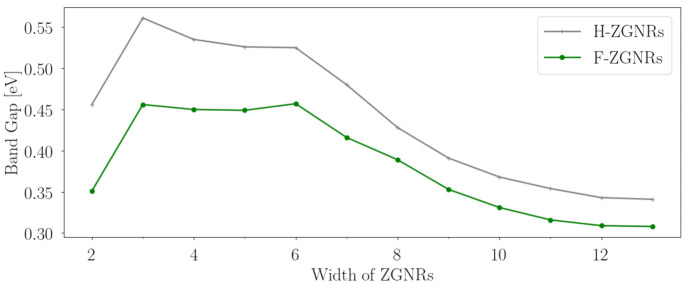
Bandgaps in eV for 2–13 widths of ZGNRs at the PBE level. For the stability, the dangling bonds at the edge are saturated with hydrogen and fluorine atoms, respectively.

**Figure 3 nanomaterials-12-00125-f003:**
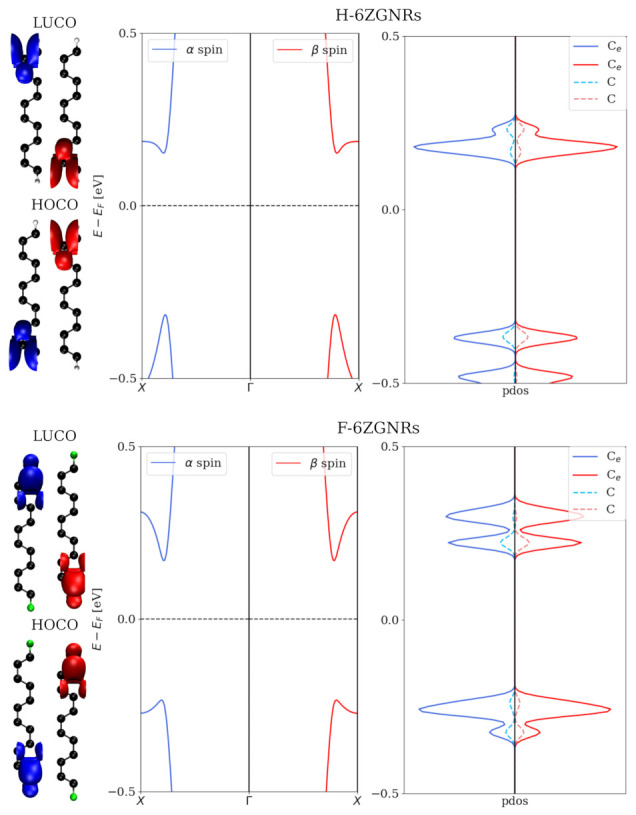
Bandstructures and projected density of states (PDOS) on two different groups of carbon atoms (C${}_{e}$ denotes edge ones, while C denotes the others) of H-6ZGNRs and F-6ZGNRs are shown. The lowest unoccupied crystal orbitals (LUCO) and highest occupied crystal orbitals (HOCO) for each structures are illustrated. The α and β spin channels are presented in red and in blue, respectively.

**Figure 4 nanomaterials-12-00125-f004:**
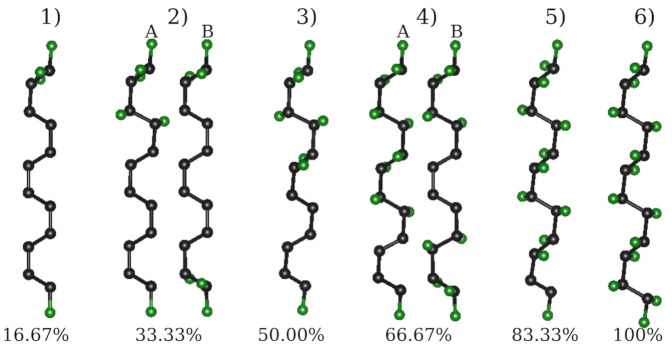
Top view of simulated fluorinated F-6ZGNRs with 6 different fluorination degrees. (A) denotes the asymmetric configuration, whereas (B) denotes the symmetric configuration.

**Figure 5 nanomaterials-12-00125-f005:**
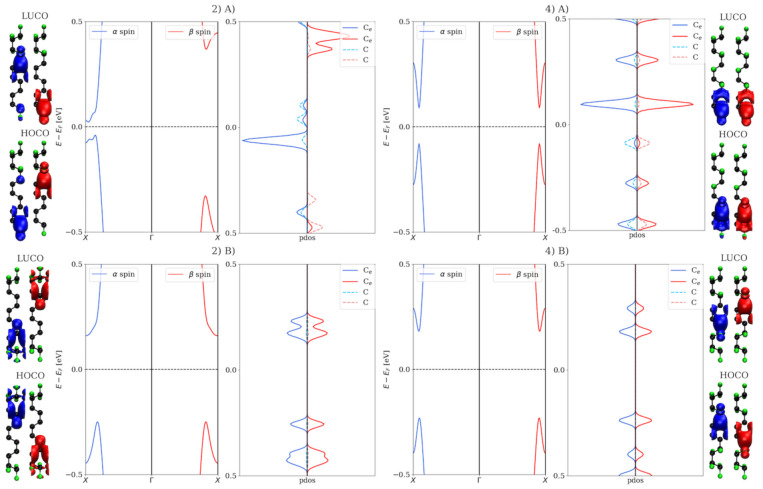
Bandstructures and projected densities of states (PDOS) on two configurations of the 33.33% and 66.67% fluorinated F-6ZGNRs are shown. The lowest unoccupied crystal orbitals (LUCO) and highest occupied crystal orbitals (HOCO) for each structures are illustrated.The α and β spin channels are presented in red and in blue, respectively.

**Figure 6 nanomaterials-12-00125-f006:**
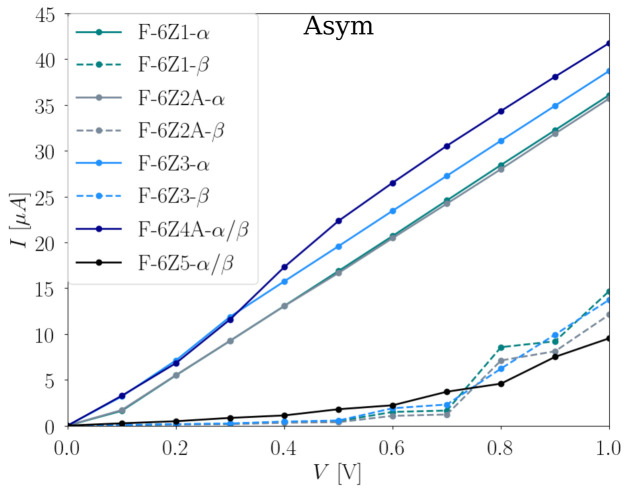
Total current I (μA) in dependence of the applied bias voltage V (V) with a 0.1 V step for the asymmetric fluorinated F-6Zs structures with five different fluorination degrees illustrated in Figure 1. It is noted that the values evaluated at each 0.1 V step are linearly connected for a better visualization of the evolution in the I with the increasing V.

**Figure 7 nanomaterials-12-00125-f007:**
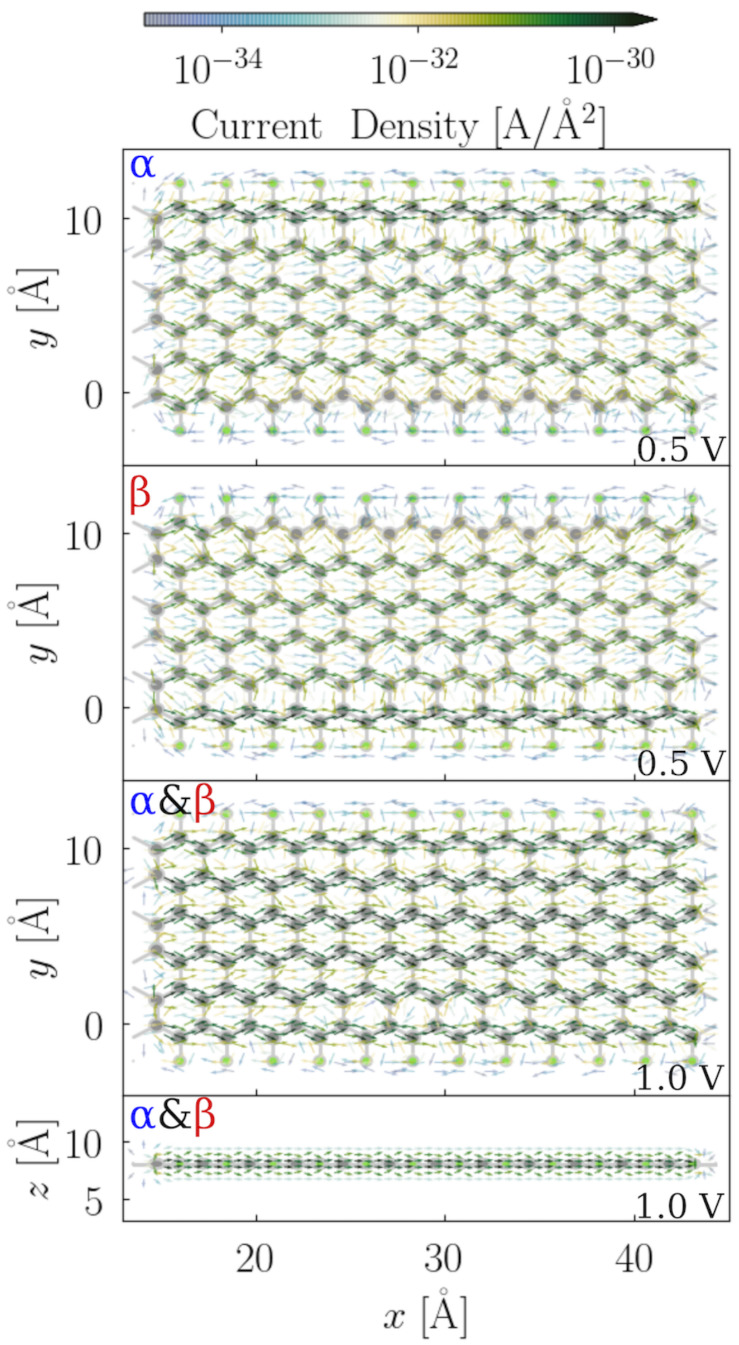
Quiver plot of the electronic current density projected on the real space of the scattering region of F-6Z. The grid spacing is chosen as 1 atomic unit in the volume of the cell cartesian coordinates. The bias voltage along the *x*-axis of the nanojunction plane is applied. The upper two panels illustrate the α and β spin channels at 0.5 V bias voltages, respectively, whereas the lowest two panels denote both α and β spin channels at 1 V. For simplification, the flux density is integrated along the *z*-axis of the nanojunction plane for the top view, and along the *y*-axis for the side view. The intensity of the current density is illustrated according to the color bar, where green indicates the high intensity and blue presents the low intensity.

**Figure 8 nanomaterials-12-00125-f008:**
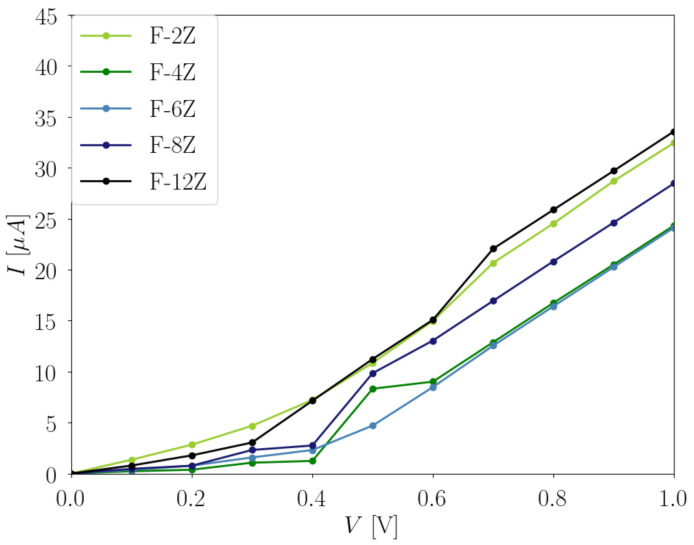
Total current I (μA) in dependence of the applied bias voltage V (V) with 0.1 V step for the F-*X*Z transport model, where X = 2, 4, 6, 8, and 12. It is noted that the values evaluated at each 0.1 V step are linearly connected for a better visualization of the evolution in the I with the increasing V.

**Figure 9 nanomaterials-12-00125-f009:**
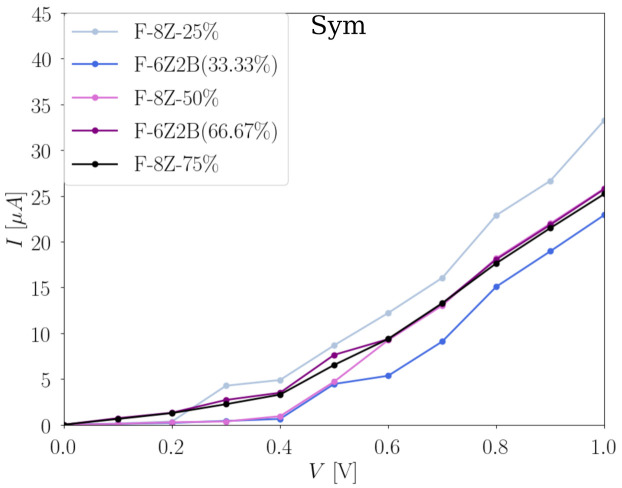
Total current I (μA) in dependence of the applied bias voltage V (V) with 0.1 V step for symmetric fluorinated F-6Zs and F-8Zs of both spin channels. It is noted that the values evaluated at each 0.1 V step are linearly connected for a better visualization of the evolution in the I with the increasing V.

**Figure 10 nanomaterials-12-00125-f010:**
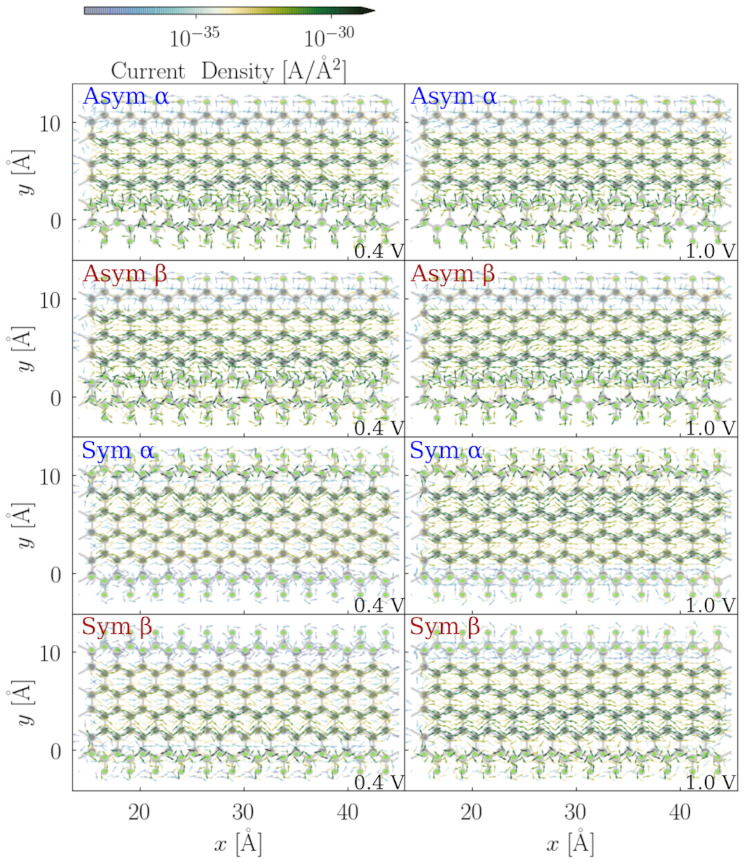
Quiver plot of the electronic current density projected on the real space of the scattering regions of both asymmetric and symmetric 33.33% fluorinated F-6Z. The grid spacing is chosen as 1 atomic unit in the volume of the cell cartesian coordinates. The bias voltage along the *x*-axis of the nanojunction plane is applied. The upper two panels illustrate the α and β spin channels of asymmetric 33.33% fluorinated F-6Z (originated from 2(A), both at 0.4 V and 1 V bias voltages, respectively, whereas the lower two panels denote the α and β spin channels of symmetric 33.33% fluorinated F-6Z (originated from 2(B)) at 0.4 V and 1 V bias voltages.

**Figure 11 nanomaterials-12-00125-f011:**
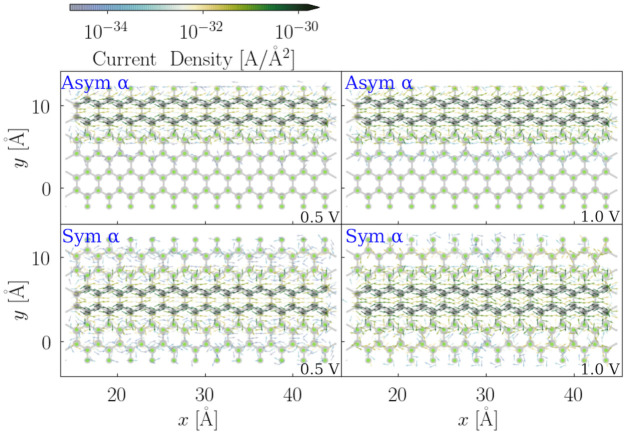
Quiver plot of the electronic current density projected on the real space of the scattering regions of both asymmetric and symmetric 66.67% fluorinated F-6Z. The grid spacing is chosen as 1 atomic unit in the volume of the cell cartesian coordinates. The bias voltage along the *x*-axis of the nanojunction plane is applied. The upper panels illustrate the α spin channel of asymmetric 66.67% fluorinated F-6Z (originated from 4(A)) both at 0.5 V and 1 V bias voltage, respectively. The lower panels denote the α spin channels of symmetric 66.67% fluorinated F-6Z (originated from 4(B)) at 0.5 V and 1 V bias voltages.

**Table 1 nanomaterials-12-00125-t001:** Bandgaps and the average binding energy per F${}_{2}$ for the most thermodynamically stable F-6ZGNR-based structures with each considered fluorination degree. All energy values are given in eV and they are obtained at the PBE level.

No.	Fluorination Degree	Bandgap		E${}_{a}$/F${}_{2}$
		α	β	
(1)	16.67%	0.006	0.785	−3.905
(2A)	33.33%	0.087	0.712	−3.756
(2B)	33.33%	0.416	0.416	−3.997
(3)	50.00%	0.092	0.686	−3.692
(4A)	66.67%	0.184	0.184	−3.677
(4B)	66.67%	0.419	0.419	−3.798
(5)	83.33%	0.645	0.645	−3.694
(6)	100.00%	3.174	3.174	−3.761

**Table 2 nanomaterials-12-00125-t002:** Bandgaps for both α and β channels, and the average binding energy per F${}_{2}$ for the most thermodynamically stable structures at each considered fluorination degree. It is noted that the bandgap of two spin channels are degenerate. All energy values are given in eV and they are obtained at the PBE level.

No.	Fluorination Degree	X-ZGNRs	Bandgap	E${}_{a}$/F${}_{2}$
(a)	33.33%	6ZGNRs	0.416	−3.997
(b)	66.67%	6ZGNRs	0.419	−3.798
(c)	100.00%	6ZGNRs	3.174	−3.761
(d)	25.00%	8ZGNRs	0.339	−3.985
(e)	50.00%	8ZGNRs	0.409	−3.804
(f)	75.00%	8ZGNRs	0.423	−3.716
(g)	100.00%	8ZGNRs	3.113	−3.697
(h)	16.67%	12ZGNRs	0.188	−3.933
(i)	33.33%	12ZGNRs	0.209	−3.776
(j)	50.00%	12ZGNRs	0.319	−3.694
(k)	66.67%	12ZGNRs	0.395	−3.657
(l)	83.33%	12ZGNRs	0.423	−3.637
(m)	100.00%	12ZGNRs	3.044	−3.630

## Data Availability

The data that support the findings of this study are available from the corresponding author upon reasonable request.

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
