# Peer review of "Edge Effect in Electronic and Transport Properties of 1D Fluorinated Graphene Materials"

_nanomaterials, 2021, doi:10.3390/nano12010125_

Round 1

Reviewer 1 Report

The article "Edge Effect in Electronic and Transport Properties of 1D Fluorinated Graphene Materials" by Jingjing Shao and Beate Paulus presents a theoretical investigation on fluorinated graphene nanoribbons. The main focus of this investigation is to draw a relation between the fluorination rate of nanoribbons and their electron transport properties. The inspiration for the present article, as stated by the authors, is an experimental study where selective fluorination patterns on graphene are realized with partial control of flurination ratio and arrangement of F atoms. The authors of the article submitted for publication model graphene with 1D structures, i.e. graphene nanoribbons with various size. Such atomic model might not be the best possible to mimick the 2D structure of graphene. However, 1D graphene nanoribbons are interesting by themselves, and the subject is of potential interest for a broad scientific community.

The theoretical method adopted by the authors is well described and well grounded in physics. The results presented in this paper are likely to be reliable.

The article is well written and organized. The quality of artworks is high.  References are appropriate.

I have just a minor suggestion to improve this article. Electron transport can be treated with a variety of theoretical approaches and computational implementations. The authors might spend few words to clearify the advantages and the shortcoming of the one adopted here, to help the reader in the comprehension of the general subject.

Overall, a very good paper.

Reviewer 2 Report

I found the results presented in this manuscript about the electron transport properties of fluorinated-graphene sheets using the NEGF method are interesting to the field of spintronics. Specifically, the calculations clearly demonstrate the possible way to control the electronic band gaps and the spin degeneracy of electron energy levels in various F-functionalized graphene nanowires. The methods and the  results are sound. The discussions are in general very clear to me.  The paper could be published after some minor revisions. My comments are below:

  1. The title of the paper is a kind of confusing to me. After I carefully going through the manuscript, the predictions strongly favor that the fluorination patterns and the degree of Fluorination play the most vital role to tune the electronic structures of F-functionalized graphene nanowires, the edge effects only modifies the band gaps, because the edge effects simply do not alter the spin degeneracy. And if the current manuscript is more targeted at the spintronics, then the edge effects is an less relevant point. Although the band gap definitely would change the response of current to the applied voltage, but that is well known theoretically even without an expensive NFGE calculations.
  2.   Figure.2 is interesting, the band  gaps showed a non-monotonical change versus the width of ZGNRs, can the author compose an physical mechanism for this behaviour?
  3. Please the expand the energy window for plotting the electron density of states in Figure. 2 or Figure.3. I understand the main point here is just show the electron energy level at or near Fermi level, because those states contribute most to the electron conductivity. However, I have noticed that the discussions about the correlations among current, electronic states and applied voltages in sessions 3.3.1 and 3.3.2, you mentioned the electronic states beyond 0.5eV could be used to explain the current versus voltage for various F-functionalized graphene nanowires at higher V.   Maybe it is also necessary to show all electronic states in a range from -1.0eV to 1.0 eV?
  4. The current versus voltage profiles for different F-graphene nanowires are difficult to be distinguished from each other in the plain printed version , please change the plot style. 
  5. The current method and implementation calculated the current flows from different channels separately, assuming no spin-flipping between different spin channels, but for spintronics, the spin dynamics and its relaxation are also important, can the authors explain why spin flipping dynamics is not considered or mentioned in the session 2. 
